# Socioeconomic status differences in psychological responses to unfair treatments: Behavioral evidence of a vicious cycle

**Youngju Kim[1]°, Jaewuk Jung[2]°, Jinkyung Na[2]\***

**1** Center for Happiness Studies, Seoul National University, Seoul, The Republic of Korea, **2** Department of Psychology, Sogang University, Seoul, The Republic of Korea

° These authors contributed equally to this work.
\* jinkyung@sogang.ac.kr

## Abstract

Two studies investigated whether lower socioeconomic status (SES) would be associated with greater tolerance for unfair treatments. Specifically, we hypothesized that individuals with lower SES would be less likely to perceive apparent injustice as unfair than those with higher SES, and furthermore, such differences in perception would lead to the corresponding differences in ensuing psychological responses. In support of the hypotheses, we found that (Study 1, $N = 326$; Study 2, $N = 130$), compared with higher SES participants, lower SES participants perceived one-sidedly disadvantageous distribution during the dictator game as less unfair. Moreover, a behavioral experiment in Study 2 showed that such tolerance for unfair treatments were associated with subsequent passive reactions in the ultimatum game. Taken together, the results imply a vicious cycle whereby the SES differences in a tendency to accept unfair treatments lead to psychological responses that may maintain or even strengthen the existing social disparities.

## Introduction

Experiencing unfair treatments is detrimental in many respects [1, 2]. Thus, it would be quite important to detect it when one is treated unfairly since such perception is the first step to acting on unfairness and avoiding further damages. In addition, such awareness is particularly important among individuals with lower socioeconomic status (SES) as they face societal/personal unfairness more often than individuals with higher SES due to financial/educational constraints, low social capital, and negative stereotypes/ discrimination [3]. Given that lower SES individuals are more likely to be treated unfairly, one may expect lower SES individuals to be less tolerant of unfairness than higher SES individuals. However, we reason that lower SES individuals, through chronic/repeated experiences with unfair treatments, would become tolerant to unfair treatments toward them. Therefore, we predicted that those most in need of sensitivity to unfair treatments, namely lower SES individuals, would perceive unfair treatments as less unfair than higher SES individuals. Furthermore, such tolerance would be

**Funding:** This work was supported by the National Research Foundation of Korea (NRF; https://www.nrf.re.kr/index) grant funded by the Korea government (KRF-2019S1A5A8032810). The grant was awarded to JN. The funders had no role in study design, data collection and analysis, decision to publish, or preparation of the manuscript.

**Competing interests:** The authors have declared that no competing interests exist.

associated with subsequent psychological responses that may lead to further unfair treatments. In the present study, we aimed to test this kind of vicious cycle across two studies.

## Socioeconomic status and unfair treatments

Numerous research generally found that individuals from lower SES backgrounds encounter social unfairness more often than individuals from higher SES backgrounds (e.g., see [3, 4]. For example, lower SES individuals are treated more unfairly than higher SES individuals in a way that can reinforce the existing social hierarchy [5, 6]. Similarly, lower SES individuals are frequent targets of negative stereotyping and discrimination [e.g., 7–9]. For example, Rivera and Tilcsik [10] found that higher SES applicants were more successful than lower SES applicants in spite of identical credentials. Moreover, those with lower SES have a number of opportunities to witness such discriminations and unfair treatments against other lower SES individuals in their immediate social network [11]. In contrast, higher SES individuals are usually treated fairly, if not favorably [e.g., 5]. Likewise, higher SES individuals only rarely witness discriminations against their fellow higher SES individuals [12]. Moreover, even when higher SES individuals are favorably treated, they may not see such treatments as unfair and even believe that they deserve better outcomes than others [13].

Since one's idiosyncratic samples of experiences shape their beliefs about social world [14], the SES differences in the frequency of unfair treatments could lead to the correspondingly different worldviews. In fact, lower SES individuals tend to have less favorable view about the world and themselves, compared with higher SES individuals [15]. For example, lower SES individuals are less optimistic regarding their future [16] and they have lower trust in other people in general [17] than higher SES individuals. Likewise, one's SES is negatively associated with their perceived power [18] and control over their situation [19]. In addition, lower SES individuals have a tendency to believe that they are not competent [20] and that their success is unlikely due to pervasive negative stereotypes [21].

Taken together, previous research suggests that lower SES individuals are more likely to face chronic/repeated exposures to unfair treatments and consequently, have relatively more negative perspectives on their life than higher SES individuals. Moreover, chronic/repeated exposures to stressful events lead to habituation and eventually, learned helplessness [22]. Thus, we reasoned that lower SES individuals would become accustomed to unfair treatments. On the other hand, higher SES is associated with entitlement [23, 24] and entitlement is shown to be linked to feelings of injustice in response to unfair treatments [25]. That is, individuals with higher SES are likely to be sensitive to unfairness due to their heightened entitlement [e.g., 25]. Based on these findings, we predict that those who are most likely to be victims of unfair treatments—namely, lower SES individuals—would perceive unfair treatments as *less* problematic than higher SES individuals. The first goal of the present research was to empirically test the proposed SES differences in tolerance for unfairness.

Our second goal was to investigate the downstream consequences of the hypothesized SES differences in one's tolerance for unfair treatments. To the extent that one believes that a given unfair treatment is less problematic (i.e., greater tolerance for unfairness), the person is less likely to take an action against the unfair treatment. Therefore, we predicted that the SES differences in perception of unfairness would be associated with the corresponding differences in the likelihood to engage in action that may improve the current unfair situation, to take an issue with the person who is responsible for unfairness, and to avoid further interaction with them. In sum, we propose that SES would be associated with psychological responses to unfair treatments through one's perception of unfairness. This proposition is in line with previous research showing that, when people are faced with unfairness, those from advantaged groups are more

likely to take action against their disadvantages [e.g., 26], whereas those from disadvantaged groups are relatively insensitive to such treatments [e.g., 25, 27]. Moreover, the current research can extend this relevant literature by showing how SES differences in initial tolerance for unfairness could have downstream consequences on subsequent psychological responses.

## The present research

The present research aimed to test the hypothesis that lower SES individuals would be more tolerant to unfair treatments than higher SES individuals and further, such differences in tolerance for unfairness would lead to *passive* psychological reactions subsequently on the part of lower SES individuals. Toward this end, we exploited two resource allocation games, namely, the dictator game (DG) and the ultimatum game (UG). These games are a well-established paradigm to implement unfairness and investigate resulting psychological reactions [e.g., 25, 28]. Both games are a two-person decision game involving a proposer and a recipient. In the DG, a proposer (i.e., a dictator) decides on a split of monetary resources. Then, a recipient remains passive and has to accept the proposer's allocation. In the UG, a proposer suggests a split of monetary resources to the recipient, similar to the DG. However, unlike the DG, the recipient can choose to either accept or reject the proposer's offer. If the recipient accepts, both players receive the proposed allocation. If the recipient rejects the offer, neither player receives anything. In other words, if the proposer decides how to distribute allocated resources between them, the recipient should accept the proposal in the DG and yet, the recipient can reject the proposal in the UG. In the present research, we used the DG to manipulate unfairness and the UG to measure one's behavioral reactions to unfairness. Specifically, in Study 1, we investigated the relationship between SES and tolerance for unfairness by measuring how lower and higher SES participants perceived a series of unfair offers they had to accept during the DG. Here, we operationalized the tolerance for unfairness in relation to the perception of unfair treatment. We reason that, if one perceives apparent unfair treatments as less unfair than others, the person can be said to be relatively tolerant of unfair treatments, compared to others.

Then, in Study 2, we conducted a behavioral experiment in order to replicate the SES differences in tolerance for unfairness and further examine the downstream consequences of such differences. Participants in Study 2 were invited to a lab and played the DG with a confederate. During the game, the confederate acting as a proposer made a series of unfair offers that the participants could not help but accepting. Then, participants played the UG with the same confederate where they could choose to accept or reject unfair offers proposed by the confederate. In this way, we could measure not only how participants' SES would influence their perception of unfair treatments during the DG but also whether and (if so) how SES differences in perceived unfairness would be associated with their behavioral responses during the UG (i.e., the acceptance of unfair offers). We also examined participants' emotional reactions, evaluation of the proposer, and intention for further interaction in Study 2.

Taken together, incorporating cognitive, affective, and behavioral measures, the present research allowed us to examine whether lower SES individuals would be more tolerant to unfair treatments than high SES individuals and how such tolerance would be associated with subsequent psychological responses. Given that greater tolerance for unfairness may contribute to maintaining the existing disparities, we believe that the present research could have important implications for psychological processes whereby social inequality is perpetuated.

## Study 1

Study 1 aimed to investigate if the lower SES individuals would perceive unfair offers as less unfair than higher SES individuals.

## Method

**Participants.** A power analyses for an independent *t* test using G*power 3.1 [29] indicated that we needed at least 64 participants for each group to detect a medium-sized (*d* = 0.50) difference with a power of .80 and an alpha-level of .05. Using this number as a reference point, we decided to recruit approximately 150 participants for each SES group due to the uncertainty regarding the actual effect size as well as potential dropouts. We recruited participants through Hankook Research, a research company in South Korea. Hankook Research has a nationwide online research panel consisting of more than three million Koreans. The panel members satisfying the following criteria were invited to take part in Study 1. In Study 1, we targeted lower and higher SES samples based on objective criteria. We used three objective criteria most frequently used in the relevant literature [e.g., 30] to recruit participants with lower vs. higher SES backgrounds: i) education level (less than BA degree vs. BA or more advanced degree), ii) income (less vs. more than the median household income in Korea), and iii) occupation (temporary/blue-collar vs. permanent/white-collar jobs). When determining participant's SES group, we checked whether their monthly household income was above (higher SES group) or below (lower SES group) the monthly median household income in South Korea for the year 2019 [31] with taking into account their household size. Using these criteria, we managed to recruit two distinctive groups of participants: 163 lower SES and 163 higher SES participants (for the detailed information, see Table 1). A sensitivity power analysis showed that our sample size of 326 participants could detect a small-to-medium size effect with a power of .80 and an alpha-level of .05.

**Procedure and materials.** Participants completed an online survey. In the survey, we first measured variables that might be associated with the perception of unfair treatments. Since one may be willing to accept apparent unfairness in order to maintain the status quo, we included one item measuring political orientation (1 = *very liberal* to 7 = *very conservative*; *M* = 3.79, *SD* = 1.17) and the eight-item system justification belief [32; Cronbach's α = .794; *M* = 3.48, *SD* = 0.83]. Similarly, we hypothesized that lower SES individuals would show a habitual/psychological tendency to tolerate unfair treatments and yet, financially insecurity itself (regardless of psychological tendencies) may force one to accept even unfair offers. Thus, we included the In-Charge Financial Distress/Financial Well-Being Scale [33] that measures the degree of financial insecurity across eight items (1 = *not at all* to 10 = *a great deal*; Cronbach's α = .927; *M* = 6.19, *SD* = 1.82) in order to (at least partially) rule out an alternative hypothesis that lower SES individuals would be more willing to accept even unfair monetary offers than higher SES individuals only because of their financial difficulty. These variables were treated as potential covariates along with demographic information (i.e., gender & age) in our analyses.

Next, participants were given a brief description about a resource allocation game, namely the dictator game (DG). Specifically, they were informed that 1) there were two players in this game, a proposer and a recipient, 2) the proposer were given 10 coins and decide how to distribute them between two players in each round, and 3) the recipient should accept the proposed offer. Then, they were asked to imagine that they were playing three rounds of the resource allocation game as a recipient. In this hypothetical gameplay, participants received lopsidedly unfair offers across three rounds. Specifically, participants received 3, 1, & 0 coins, whereas the proposer kept 7, 9, & 10 coins in the first, second, and third round, respectively (i.e., 1R = 3:7, 2R = 1:9, & 3R = 0:10). To emphasize unfairness, participants were informed that the proposer obtained 26 (7+9+10) coins, whereas they obtained a total of 4 (3+1+0) coins across the three rounds. Following the summary, participants indicated how unfair the offers from the proposer were (1 = *not at all* to 7 = *a great deal*; *M* = 5.75, *SD* = 1.34) as an index of

**Table 1. Demographic information in Study 1.**

| Category | Higher SES group (Percentage) | Lower SES group (Percentage) |
|---|---|---|
| **Gender** | | |
| Women | 50.3% | 50.9% |
| Men | 49.7% | 49.1% |
| **Age** | | |
| $M_{age}$ | 45.71 | 46.10 |
| $SD_{age}$ | 7.58 | 7.84 |
| **Education** | | |
| Junior high school | 0% | 1.8 |
| High school | 0% | 67.5 |
| Associate Degree | 0% | 27.6 |
| Some university without degree | 0% | 3.1 |
| University with degree | 76.1% | 0% |
| Graduate school (Masters/PhD) | 23.9% | 0% |
| **Family monthly income** (actual income) | | |
| $M_{income}$ | 6,410,100 (KRW) | 2,766,600 (KRW) |
| $SD_{income}$ | 2,426,890 | 1,235,810 |
| **Occupation** | | |
| Professionals (e.g., lawyers, MDs, etc.) | 13.5% | 0% |
| Managers | 12.9% | 0% |
| Office worker | 73.6% | 0% |
| Temporary service and sales workers | 0% | 45.4% |
| Temporary blue-collar worker (e.g., construction/ factory workers) | 0% | 39.9% |
| Other temporary workers | 0% | 14.7% |
| **Subjective SES** (self-reported) | | |
| $M_{subjective\ SES}$ | 3.50 | 2.40 |
| $SD_{subjective\ SES}$ | 0.69 | 0.82 |
| $N$ | 163 | 163 |

tolerance for unfair treatments. Participants next indicated which of the following SES categories they currently belonged to relative to the others in their society [e.g., 34]: the lowest class (coded as "1"), lower class, lower middle class, upper middle class, upper class, or the highest class (coded as "6"). Using this subjective measure of SES, we examined whether our division of higher SES vs. lower SES groups were valid. Finally, they reported gender and age and were debriefed.

All materials, analyzed data, and codes used in the present research are available at https://osf.io/bpfea/?view_only = fc915f0d15fe448cbed45685c6d3e61e and the present research was approved by the Institutional Review Board at Sogang University (SGUIRB-A-1909-35).

## Results

**Subjective SES.** Participants in the lower SES group ($M = 2.40$, $SD = 0.82$) indicated that they belonged to lower SES categories than participants in the higher SES group ($M = 3.50$, $SD = 0.69$), difference = -1.098, 95% CI [-1.263, -0.933], $t$ (314.48) = -13.09, $p < .001$, Cohen's $d = 1.45$. This suggest that our criteria (i.e., education, income, & occupation) were reasonably valid in creating two groups that were distinctive in terms of their SES backgrounds.

**Correlations.** Binary correlations were summarized in Table 2. SES group was positively correlated with perceived unfairness during the DG ($r = .110$, $p = .047$), indicating that participants in the lower SES group were less likely to indicate that the distribution during the DG was unfair. Note that neither demographic nor the other control variables (in particular, financial insecurity) correlated with participants' perception of unfair treatments (all $p$-values were $\geq .104$).

**SES and perceived unfairness.** Participants in the lower SES group were less likely to indicate that the distribution during the DG was unfair ($M = 5.61$, $SD = 1.34$) than participants in the higher SES group ($M = 5.90$, $SD = 1.33$), difference = -0.295, 95% CI [-0.585, -0.004], $t$ (323.994) = -2.00, $p = .047$, Cohen's $d = 0.22$. This association remained significant controlling for gender, age, political orientation, system justification belief, and financial insecurity, $F$ (1, 319) = 6.83, $p = .009$, $\eta_p^2 = .021$. In other words, consistent with our prediction, participants with lower SES were more tolerant to apparent unfair treatments than participants with higher SES. Among the three SES criteria, family income was positively correlated with perceived unfairness ($r = .118$, $p = .032$). We repeated the same analysis using income as an indicator of SES and reached the same conclusion (see S5 Table in S1 File).

## Study 2

Study 1 provided initial support to our hypothesis that lower SES participants would be less likely to perceive unfair treatment as unfair compared with their higher SES counterparts. Study 2 sought to replicate the association between SES and tolerance for unfairness and further examine its downstream consequences in a behavioral experiment. Also, Study 2 included emotional reactions to unfair treatments as an additional measure of tolerance for unfair treatments. Finally, whereas we targeted lower and higher SES samples based on objective criteria in Study 1, we measured both objective and subjective SES and treated them as a continuous variable.

### Method

**Participants.** We recruited participants via posting on an online bulletin board at a university in South Korea. Since we conducted the experiment in the midst of the COVID-19 pandemic, we did not pre-determine the number of participants. Rather we decided to recruit as many participants as possible during the study period. One hundred sixty-two graduate and undergraduate students participated in the experiment in return for an online gift voucher

**Table 2. Study 1: Correlations between all measures ($N = 326$).**

| Variables | 1 | 2 | 3 | 4 | 5 | 6 | 7 |
|---|---|---|---|---|---|---|---|
| 1. SES group | – | | | | | | |
| 2. Perceived unfairness | .110* [.001, .216] | – | | | | | |
| 3. Gender | -.006 [-.115, .103] | -.030 [-.138, .080] | – | | | | |
| 4. Age | -.026 [-.134, .083] | -.045 [-.152, .064] | .046 [-.063, .154] | – | | | |
| 5. Political orientation | -.018 [-.127, .090] | -.002 [-.110, .107] | .078 [-.031, .185] | .037 [-.072, .145] | – | | |
| 6. System justification belief | .118* [.010, .224] | -.090 [-.197, .019] | -.101 [-.208, .007] | .061 [-.048, .168] | -.023 [-.131, .086] | – | |
| 7. Financial insecurity | -.399*** [-.486, -.303] | .068 [-.041, .175] | -.146** [-.251, -.038] | -.077 [-.184, .032] | -.022 [-.131, .087] | -.357*** [-.448, -.258] | – |

*Notes*. SES group: 0 = lower SES group & 1 = higher SES group; Gender: 0 = men & 1 = women; The 95% confidence intervals are in bracket;

* $p < .05$,

** $p < .01$,

*** $p < .001$.

worth approximately 5 USD. Thirty-two of them were excluded due to their familiarity with both resource allocation games (i.e., DG & UG) since previous knowledge about these games is shown to systematically influence one's responses such that knowledgeable participants tend to accept even the smallest possible reward in the UG as it is considered as rational from an economic perspective [e.g., 35]. The inclusion of these participants did not change the direction of the results although the effects became substantially weaker. The final sample size included 130 participants (71 females; $M_{age}$ = 23.06 years, $SD_{age}$ = 2.68), which allow detection of a medium-size effect ($d$ = 0.50) with 80% power.

**Procedure and materials.** The overview of the experimental procedure is shown in Fig 1A. Upon arriving at the laboratory, participants were seated in an individual cubicle and informed that they would play two different types of resource allocation games with another participant (in reality, a confederate) via an online messenger application (i.e., LINE; https://line.me/en/). Then, we measured their baseline emotions by asking them to indicate how they were currently experiencing each of the following four emotions such as sadness, anger, calm, and pleasant on a 7-point scale (1 = *not at all* to 7 = *a great deal*). Next, participants were informed that their goal in this experiment was to obtain as many coins as possible during the two games and they could get an additional reward (i.e., online gift voucher worth approximately 10 USD) based on their performance.

*The dictator game.* Participants were invited to an online chatting room and met another participant, namely the confederate (Fig 1B). Then, we instructed that the proposer in this game would be given ten coins in each round and decide how to distribute the coins between two players, whereas the recipient would have no other option to accept the proposed offer. Participants were led to believe that they were randomly assigned to the recipient role. After probing their understanding of the game, they played three rounds of the DG. As in Study 1,

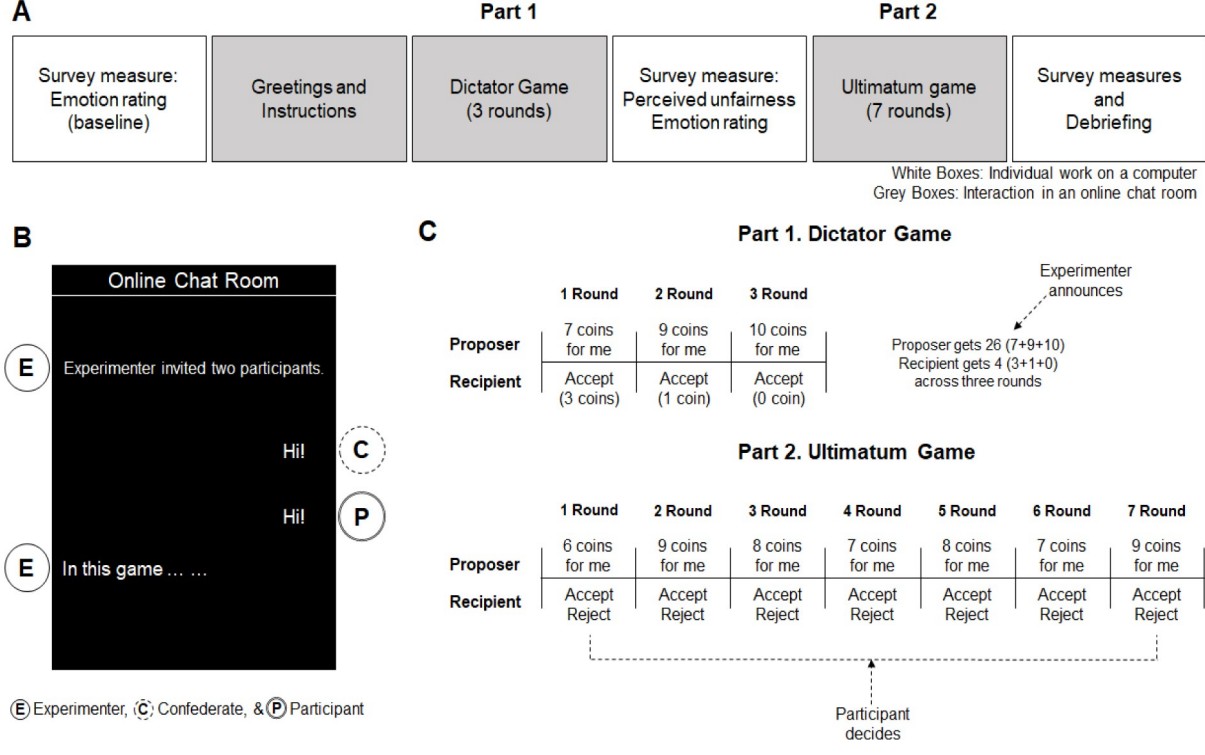

**Fig 1. Experimental procedures in Study 2.**

the proposer made apparently unfair proposals across three rounds (Fig 1C). Again, the experimenter announced that, across three rounds, the proposer obtained 26 coins and the recipient obtained 4 coins at the end of the DG. Then, participants completed an online survey. In the survey, participants reported how unfair the offers from the proposer were (1 = *not at all* to 7 = *a great deal*; $M = 6.27$, $SD = 1.02$). For emotional responses, participants reported their emotions after the DG in the same way as at the baseline. We averaged their emotional responses to the four items after reverse-coding positive affects (i.e., calm & pleasant; Cronbach' $\alpha$s = .625 & .650, $M$s = 3.28 & 4.30, $SD$s = 0.92 & 1.05 at the baseline and after the DG, respectively). Higher values in this index reflect more negative emotional experiences.

*The ultimatum game.* After individually working on the online survey, participants were brought back to the chatting room and told that they would play another game with the same partner. Participants were instructed that the proposer (i.e., the confederate) would decide how to distribute the 10 coins in each round as in the previous game. However, we emphasized that, unlike in the previous game, the recipient (i.e., participants) could either accept or reject the offered proposal. They were also informed that, when rejecting an offer in a given round, both the proposer and the recipient would get zero coin in that round. They played seven rounds of the UG and, as in the DG, the offers made by the proposer were always unfair to participants (see Fig 1C). The participants indicated whether to accept or reject the proposed unfair offers in each round and we counted how many times participants accepted unfair offers across seven rounds and used it as behavioral indicator of passive reaction to unfair treatment ($M = 2.99$, $SD = 2.25$).

*Follow-up survey.* Next, participants were asked to fill out an online follow-up survey. First, participants were asked to evaluate the game partner with six adjectives (i.e., righteous, smart, reliable, competent, generous, knowledgeable) on a 7-point scale (1 = *not at all* to 7 = *a great deal*). Their responses were averaged to form an index of positive evaluation of the unfair proposer (Cronbach's $\alpha$ = .870; $M = 2.81$, $SD = 1.13$). Participants also indicated how much they would be willing to play the resource allocation games again as a recipient role with the same partner in the future on a 7-point scale (1 = *definitely will not* to 7 = *definitely will*; $M = 2.90$, $SD = 1.65$).

The follow-up survey also included measures of our independent variable: socioeconomic status (SES). First, we measured participants' subjective perception of their SES using the same scales as Study 1 ($M = 3.77$, $SD = 0.73$). Second, we measured their objective SES with parental education and family income. For parental education, we separately measured paternal and maternal education with a 7-point scale (1 = *no formal education* to 7 = *master's degree or higher*; $M_{father} = 5.76$, $SD = 0.98$; $M_{mother} = 5.39$, $SD = 1.07$) and then, averaged these two measures to form an index of parental education ($M = 5.58$, $SD = 0.89$). For family income, participants indicated their annual household income (1 = *less than 15 million KRW* to 7 = *more than 100 million KRW*; $M = 5.16$, $SD = 1.47$). Finally, we standardized and then averaged parental education and annual household income ($r = .27$, $p = .002$) to yield our index of objective SES ($M = 0.00$, $SD = .80$).

We also measured the three control variables using the same scales as Study 1: political orientation ($M = 4.05$, $SD = 1.26$), system justification belief (Cronbach's $\alpha$ = .715; $M = 3.93$, $SD = 0.74$), and financial insecurity (Cronbach's $\alpha$ = .910; $M = 4.81$, $SD = 1.64$). As in Study 1, these variables were treated as potential covariates along with demographic information (i.e., gender & age) in the main analyses.

## Results

**Correlations.** Binary correlations were reported in Table 3. First of all, subjective SES positively correlated with perceived unfairness during the DG, $r = .209$, $p = .017$. Objective SES

**Table 3. Study 2: Correlations between all measures (N = 130).**

| Variables | 1 | 2 | 3 | 4 | 5 | 6 | 7 | 8 | 9 | 10 | 11 | 12 | 13 |
|---|---|---|---|---|---|---|---|---|---|---|---|---|---|
| 1. Subjective SES | – | | | | | | | | | | | | |
| 2. Objective SES | .610*** [.488, .707] | – | | | | | | | | | | | |
| 3. Perceived unfairness | .209* [.037, .367] | .161 [–.012, .323] | – | | | | | | | | | | |
| 4. Emotion at the baseline | –.156 [–.319, .017] | –.082 [–.251, .091] | –.038 [–.209, .135] | – | | | | | | | | | |
| 5. Emotion after the DG | .033 [–.140, .204] | .068 [–.105, .237] | .210* [.038, .368] | .280** [.112, .430] | – | | | | | | | | |
| 6. Number of acceptances of unfair offers | .103 [–.071, .270] | .117 [–.056, .283] | –.182* [–.343, –.010] | .082 [–.092, .250] | –.110 [–.277, .063] | – | | | | | | | |
| 7. Evaluation of the proposer | .037 [–.136, .208] | .012 [–.160, .184] | –.182* [–.343, –.009] | .153 [–.021, .316] | –.245** [–.400, –.075] | .435*** [.282, .564] | – | | | | | | |
| 8. Intention for future interaction | .032 [–.141, .203] | –.010 [–.182, .162] | –.326*** [–.471, –.162] | .009 [–.163, .181] | –.281** [–.431, –.113] | .327*** [.163, .471] | .507*** [.365, .623] | – | | | | | |
| 9. Gender | –.055 [–.225, .118] | .023 [–.149, .195] | .059 [–.114, .229] | .147 [–.026, .311] | .025 [–.148, .196] | –.058 [–.228, .115] | .017 [–.156, .188] | .095 [–.079, .263] | – | | | | |
| 10. Age | .055 [–.119, .225] | –.026 [–.197, .147] | –.222* [–.379, –.051] | .045 [–.128, .215] | –.007 [–.179, .166] | .026 [–.147, .197] | –.044 [–.214, .129] | .008 [–.164, .180] | –.176* [–.337, –.003] | – | | | |
| 11. Political orientation | –.089 [–.257, .085] | –.022 [–.193, .151] | –.004 [–.176, .169] | .007 [–.165, .179] | .048 [–.125, .218] | .041 [–.132, .212] | –.034 [–.205, .139] | –.031 [–.202, .142] | –.102 [–.269, .072] | .162 [–.011, .324] | – | | |
| 12. System justification belief | .261** [.092, .414] | .229** [.058, .385] | .037 [–.136, .208] | –.327*** [–.471, –.162] | –.211* [–.369, –.040] | .051 [–.122, .221] | .055 [–.118, .225] | .085 [–.089, .253] | –.061 [–.230, .113] | .039 [–.134, .210] | .065 [–.109, .234] | – | |
| 13. Financial insecurity | –.409*** [–.541, –.253] | –.404*** [–.537, –.247] | –.031 [–.202, .142] | .394*** [.237, .529] | .232** [.061, .388] | .038 [–.135, .209] | .084 [–.090, .252] | –.139 [–.304, .034] | .022 [–.150, .194] | .067 [–.107, .236] | –.071 [–.240, .103] | –.494*** [–.613, –.350] | – |

*Notes.* Gender: 0 = men & 1 = women; The 95% confidence intervals are in bracket;

* $p < .05$,

** $p < .01$,

*** $p < .001$.

showed the same pattern although the correlation was "marginally" significant, $r$ = .161, $p$ = .068. That is, consistent with our hypothesis lower SES participants were less likely to indicate that the distribution during the DG was unfair. As also expected, perceived unfairness negatively correlated with the number of accepted offers during the UG ($r$ = -.182, $p$ = .038), evaluation of the proposer ($r$ = -.182, $p$ = .038), and intention for future interaction ($r$ = -.326, $p$ < .001). In other words, to the extent that one perceived the unequal distribution during the DG as relatively less unfair, the person accepted unfair offers more frequently, evaluated the proposer more positively, and were more willing to play the games in the same way. In this sense, it can be said that one's perception of unfair treatments was associated with passive responses. Curiously, neither subjective nor objective SES significantly correlated with emotions after the DG, although subjective SES showed a "marginally" significant association with emotions at the baseline, $r$ = -.156, $p$ = .077, indicating that higher SES was associated with more positive emotions at the baseline. Finally, neither demographic nor the other control variables correlated with participants' perception of unfair treatments, except for age ($r$ = -.222, $p$ = .011).

We note that subjective and objective SES showed an identical pattern, although the results were somewhat weaker when using objective SES than when using subjective SES as a predictor. Thus, we reported the results based on subjective SES in the main text below and the results based on objective SES were included in the (see S1 Table and S2 Fig and S3 Table in S1 File).

**SES and perceived unfairness.** Then, we examined how subjective SES would be associated with one's perception of unfair treatments during the DG. Specifically, perceived unfairness was regressed on subjective SES without (Model 1) or with (Model 2) five covariates. The full results are summarized in Table 4. As expected, we found that subjective SES was positively associated with perceived unfairness, $b$ = 0.291, $SE$ = 0.120, $p$ = .017, 95% CI [0.053, 0.528], indicating that lower SES individuals tended to be more tolerant to apparent unfair distributions during the DG. Such association remained significant, even when gender, age, political orientation, system justification belief, and financial insecurity were held constant, $b$ = 0.376, $SE$ = 0.132, $p$ = .005, 95% CI [0.115, 0.637]. To sum up, Study 2 successfully replicated the findings in Study 1 with a continuous measure of SES.

**SES and emotional responses.** Next, we tested how emotional responses toward the unfair treatments after the DG would vary as a function of one's subjective SES. We found that, compared with emotions at the baseline ($M$ = 3.28, $SD$ = 0.92), emotions after the DG

**Table 4. Study 2: The results of the multiple regression analysis testing the effects of SES on perceived unfairness during the dictator game.**

| Predictors | Perceived unfairness during the dictator game | | | | | |
|---|---|---|---|---|---|---|
| | Model 1 | | | Model 2 | | |
| | B | SE | 95% CI | B | SE | 95% CI |
| SES | 0.291* | .120 | [0.053, 0.529] | 0.376** | .132 | [0.115, 0.637] |
| Gender | | | | 0.074 | .177 | [−0.275, 0.424] |
| Age | | | | −0.095** | .034 | [−0.162, −0.029] |
| Political orientation | | | | 0.057 | .071 | [−0.083, 0.197] |
| System justification belief | | | | 0.043 | .136 | [−0.226, 0.313] |
| Financial insecurity | | | | 0.072 | .065 | [−0.057, 0.202] |
| adjusted $R^2$ | .036 | | | .068 | | |

*Notes.* Unstandardized coefficients are given; CI = confidence interval for B; Gender: 0 = men & 1 = women;

\* $p$ < .05,

\*\* $p$ < .01.

($M$ = 4.30, $SD$ = 1.05) became significantly more negative, $F$ (1, 125) = 17.34, $p$ < .001, $\eta_p^2$ = .122. More importantly, such effect was not qualified by subjective SES, $F$ (4, 125) = 0.73, $p$ = .572, $\eta_p^2$ = .023. That is, regardless of their SES, participants showed negative emotional experiences in response to unfair distribution during the DG. We also note that SES was not significantly associated with emotions after the DG with or without controlling for emotions at the baseline, with: $b$ = 0.112, $SE$ = 0.123, $p$ = .363, 95% CI [-0.131, 0.355] & without: $b$ = 0.047, $SE$ = 0.126, $p$ = .710, 95% CI [-0.203, 0.297]. It is quite interesting that participants' report of perceived unfairness varied as a function of SES and yet, SES did not show any significant effect on their emotional responses. We will address this intriguing dissociation in the discussion.

**Behavioral and interpersonal outcomes of SES differences in perceived unfairness.** So far, we found that one's tendency to see unfairness out of apparent unfair treatments was attenuated among lower SES participants. In this section, we addressed the most important questions in Study 2 regarding behavioral and interpersonal consequences of the SES differences in perceived unfairness. Specifically, we tested whether subjective SES was indirectly associated with passive reactions to unfair situations (i.e., the number of acceptances of unfair offers, evaluation of the proposer, and intention for future interaction) through perceived unfairness. For each of our three outcome variables, we built two mediation models: without or with five covariates. Indirect effects were probed using the PROCESS macro with bootstrapping for 10,000 resamples and 95% confidence intervals [36]. In these models, perceived unfairness was served as a mediator since we are interested in whether one's perception of unfair treatments during the DG would be associated with psychological responses in the subsequent interactions.

Fig 2 presents the results of the mediation analyses. First, the indirect effect from subjective SES to the number of accepted offers during the UG through perceived unfairness was significant, indirect effect $b$ = -0.137, $SE$ = 0.083, 95% CI [-0.329, -0.013]. This indirect effect remained significant even when gender, age, political orientation, system justification belief, and financial insecurity were held constant: indirect effect $b$ = -0.197, $SE$ = 0.100, 95% CI

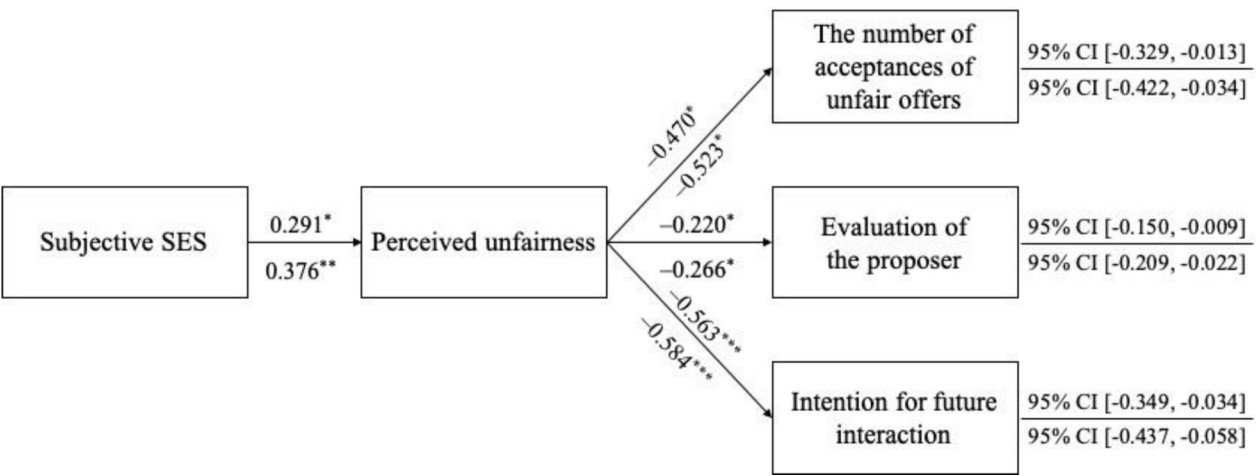

**Fig 2. Study 2: The results of the mediation analysis testing the indirect effects of subjective SES via perceived unfairness on subsequent psychological responses.** Unstandardized coefficients are given above (without covariates models) and below (with covariates models) the arrow/ lines; 95% CI = 95% bootstrap confidence interval for indirect effects (Bootstrap Sample = 10,000); Total and direct effects parameters are reported in S4 Table in S1 File; * $p$ < .05, ** $p$ < .01, *** $p$ < .001.

[-0.422, -0.034]. Essentially the same pattern observed for evaluation of the proposer, with covariates: indirect effect $b$ = -0.100, $SE$ = 0.048, 95% CI [-0.209, -0.022] & without covariates: indirect effect $b$ = -0.064, $SE$ = 0.036, 95% CI [-0.150, -0.009], and intention for future interaction, with covariates: indirect effect $b$ = -0.220, $SE$ = 0.096, 95% CI [-0.437, -0.058] & without covariates: indirect effect $b$ = -0.164, $SE$ = 0.081, 95% CI [-0.349, -0.034]. We also note that the total and direct effects of subjective SES on DVs were mostly nonsignificant. In other words, subjective SES itself was not significantly associated with DVs (see S4 Table in S1 File for detailed results).

To sum up, compared with higher SES participants, lower SES participants were likely to perceive the unequal distribution during the DG as less unfair and this tendency, in turn, was associated with passive reactions during the UG in terms of acceptances of unfair offers, evaluation of the proposer, and intention for future interaction.

## Discussion

The present research investigated and found how socioeconomic status (SES) is associated with one's responses to unfair treatments where they were the victim. First, we found that lower SES individuals are likely to perceive unfair distribution as less unfair than higher SES individuals (Studies 1 & 2). Specifically, when offered one-sidedly disadvantageous proposals during the dictator game, participants with lower SES were less likely to indicate that the proposals were unfair than their higher SES counterparts. The finding is consistent with previous research showing that those from disadvantaged groups tend to be insensitive to unfair treatments [e.g., 25, 26]. Likewise, it is also in line with previous demonstration that power and wealth are positively associated with entitlement [23, 24].

A substantial body of literature on fairness and equality has demonstrated that individuals generally show inequality aversion [e.g., 37], and thus most people reject unfairness even at the expense of personal resources [38]. Moreover, since disadvantageous people are more likely to be on the receiving end of societal injustice (compared with advantageous people), lower SES people have a greater incentive to be sensitive (and react) to the unfair situation. However, across two studies, we found that lower SES participants perceived unfair treatment as less unfair compared with higher SES participants. These findings may indicate that the lower (vs. higher) SES individuals have a different baseline of unfairness/fairness such that lower SES individuals are more tolerant of unfairness than higher SES individuals.

More importantly, we also showed the downstream consequences of the SES differences in perception of unfairness across diverse domains (Study 2). Behaviorally, we found that perceived unfairness during the dictator game was associated with the number of acceptance of unfair offers during the ultimatum game. More specifically, lower SES was associated with more acceptance of unfair proposals during the UG indirectly through the tolerance for unfairness during the DG. Not only that, we observed converging results with respect to interpersonal outcomes. Via one's perception of unfair treatments, lower SES was associated with positive evaluation of the unfair proposers and also with an intention to play the same games with the same partner and role. In other words, individuals with lower SES tend to perceive disadvantageous offers as unfair less than those with higher SES and such tolerance for unfairness is associated with psychological reactions that may invite further unfair treatments.

Given this convergence, our results in Study 2 regarding emotional responses are quite noteworthy. Specifically, participants showed negative emotional responses during the dictator game, regardless of their SES. Moreover, as shown in Table 3, such negative emotional responses correlated with more negative evaluation of the proposer and weaker intention for future interaction. These findings may suggest that lower SES individuals do feel bad about

unfair treatments and yet, they may not be able to act on such emotional reactions cognitively and behaviorally, unlike higher SES individuals. We think that it is worthwhile to further investigate this kind of divergence in psychological reactions to unfairness.

In the present research, we included several measures of SES covering both subjective and objective aspects of SES. Specifically, participants indicated educational attainment (personal and parental education in Studies 1 & 2, respectively), family income, and subjective perception of their SES. Thus, we conducted additional analyses in order to explore which indicator of SES would show a particularly strong association with perceived unfairness. In Study 1, income was positively associated with perceived unfairness, $r = .118$, $p = .032$ and the association became marginal for educational attainment, $r = .100$, $p = .071$. Curiously, subjective SES was not correlated with perceived unfairness, $r = .036$, $p = .512$. Thus, the results in Study 1 are somewhat inconsistent with the previous studies suggesting that subjective construal of one's rank in the social hierarchy would be a more powerful predictor of social cognitive tendencies than objective indices [e.g., 39]. However, in our opinion, such inconsistency should be cautiously interpreted. First, Study 1 recruited two groups that were quite distinctive in terms of their SES (i.e., education, income, and occupation). Thus, it may not be appropriate to treat SES as a continuous variable as in the correlational analyses. Rather, it may be better to conduct a group comparison as we reported in the main results. Second and more importantly, the results were not replicated in Study 2. In Study 2, subjective SES showed a stronger association than objective SES (i.e., income and parental education) as in the previous literature, Subjective SES: $r = .209$, $p = .017$, Income: $r = .142$, $p = .107$, & Parental education: $r = .114$, $p = .198$. Therefore, we think that future research should systematically investigate the potential differences between subjective and objective SES in psychological processes reported in the present research.

Although the present research found the predicted relation between SES and tolerance for unfair treatments across diverse samples (a nationally representative online panel in Study 1 & college students in Study 2) and measures of SES (education, income, occupation, and subjective rank), our data cannot speak to causal influences of SES since we did not manipulate one's SES. Likewise, although lower SES was associated with passive responses (e.g., acceptance of unfair offers during the UG) indirectly through perception of unfairness, the direct and total effects of SES were mostly nonsignificant (see S4 Table in S1 File). This suggests that SES may not be inherently associated with passive reactions. Rather, SES may be associated with various factors that have dynamic relations with passive reactions. Thus, future research should investigate other factors (than perception of unfair treatments) that play an important role in the relation between SES and psychological responses to unfairness. Additionally, our participants may not be diverse enough. For example, participants in Study 2 were college students and thus, there may not be sufficient variations in SES among them. Although we recruited non-college samples and found similar results in Study 1, all of our participants were Koreans. Therefore, the generalizability of the current findings should be further investigated in other cultural contexts with more demographic diversity.

Despite these limitations, we show that SES is associated with one's tolerance for unfair treatments and further, such association has implications for behavioral and cognitive responses that may maintain or even strengthen social disparities. In this sense, the present research can join a growing body of research showing how micro-level analyses of psychological processes can inform macro-level analyses on stratification [e.g., 40]. In conclusion, what the present research essentially highlights is a vicious cycle such that social disparities may reproduce themselves through SES differences in psychological processes resulting from the very disparities.

## Supporting information

**S1 File.**
(DOCX)

## Author Contributions

**Conceptualization:** Youngju Kim, Jinkyung Na.

**Data curation:** Youngju Kim, Jaewuk Jung.

**Formal analysis:** Youngju Kim, Jaewuk Jung.

**Funding acquisition:** Jinkyung Na.

**Investigation:** Youngju Kim, Jaewuk Jung.

**Methodology:** Youngju Kim, Jinkyung Na.

**Project administration:** Jaewuk Jung.

**Resources:** Jinkyung Na.

**Supervision:** Jinkyung Na.

**Writing – original draft:** Youngju Kim, Jaewuk Jung, Jinkyung Na.

**Writing – review & editing:** Youngju Kim, Jaewuk Jung, Jinkyung Na.

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
