## [Decision Letter · Decision Letter 0]

11 Mar 2022

PONE-D-22-04413Socioeconomic status differences in psychological responses to unfair treatments:

Behavioral evidence of a vicious cyclePLOS ONE

Dear Dr. Na,

Thank you for submitting your manuscript to PLOS ONE. After careful consideration, we feel that it has merit but does not fully meet PLOS ONE’s publication criteria as it currently stands. Therefore, we invite you to submit a revised version of the manuscript that addresses the points raised during the review process.

I have asked two experts in this area to review this manuscript and have independently read and evaluated it myself. On the whole, all of us think that the topic is very interesting and the findings have the potential to be impactful and important. However, there is still work to be done to ensure that the manuscript meets the requirements of publication at PLOS ONE. I encourage you to carefully read the reviews from both reviewers, but I will summarize some of their key concerns as well as add some of my own here. First, I find the SES categorization strange. More parsimonious would be to look just at income, which, in S1, does correlate with perceived unfairness (making lines 193-194 incorrect. Income correlated positively with perceived unfairness (r = .118, p = .032). Why not just use this basic analysis. It seems very robust to covariate inclusion and doesn’t involve any subjective judgments about what constitutes high vs. low SES.  Related to this, I’m surprised that subjective SES doesn’t correlate with the DV of interest. Typically, in psychology, we find that subjective assessments of dimensions like wealth or SES have a great influence on judgments and behavior than objective dimensions do. Why doy ou believe that, in this case, that doesn’t hold? Line 225 "The inclusion of 225 these participants did not change the direction of the results.” This statement, though true, is pretty misleading. In fact, the critical relationship between perceived SES and fairness drops well below the conventional levels of significance when including all participants. Indeed, for the excluded participants, there is a NEGATIVE relationship between SES and fairness (r = -.29, p = .10). It’s not clear to me why you would expect to see this, nor why this exclusion criteria is justified. Mere familiarity with the games shouldn't fundamentally flip the relationship between the key IV and DV. What rationale do you provide for this and was this exclusion criteria pre=determined and pre-registered, or was it made once the results were known?

In Study 2, like with S1, I’m surprised that actual income wasn’t used as a key predictor of fairness. In this study, family income is only weakly correlated with fairness (r = .14, p - .11) for the sub-sample and completely uncorrelated for the full sample (r = .07, p = .36). Again, this feels like a much more natural IV, and yet it is largely ignored.

Beyond this, there the review team feels as though there are valid questions about the use of covariates (R1), the structure of the results for S2 can be cleaned up (R1), there can be more reporting of key statistics (R1 and R2), there is confusion about the results for S2 (specifically as it relates to Figure 2 and Table 3; R1), there is concern about the use of exclusively students in S2 (R1), and, overall, the language throughout the paper could use some more precision (R1 and R1).

This is a lot to consider when working on your revision, but I am hopeful that if you can respond to these concerns with additional analysis and/or exposition, you have a good path to publication here. 

We look forward to receiving your revised manuscript.

Kind regards,

Jeff Galak, PhD

Academic Editor

PLOS ONE

Journal Requirements:

2. Peer review at PLOS ONE is not double-blinded (https://journals.plos.org/plosone/s/editorial-and-peer-review-process). For this reason, authors should include in the revised manuscript all the information removed for blind review.

Reviewers' comments:

Reviewer's Responses to Questions

**Comments to the Author**

1. Is the manuscript technically sound, and do the data support the conclusions?

Reviewer #1: No

Reviewer #2: Yes

2. Has the statistical analysis been performed appropriately and rigorously? 

Reviewer #1: No

Reviewer #2: Yes

3. Have the authors made all data underlying the findings in their manuscript fully available?

Reviewer #1: Yes

Reviewer #2: Yes

4. Is the manuscript presented in an intelligible fashion and written in standard English?

Reviewer #1: Yes

Reviewer #2: Yes

5. Review Comments to the Author

Reviewer #1: Fascinating study on how low SES is habituated to unfair treatment and how that habituation causes a vicious circle in the disparaties.

Major comments

1. The Introduction (page 3, line 37, 38) rather strongly reports that lower SES is more likely to be treated unfairly. From line 46 onwards, it is explained a bit further. Although I agree, but I still think that the unfair treatment should be explained further? How does the unfair treatment look like exactly? Perhaps some readers might not know why some treatments are unfair and why low SES receive these more often. What people consider unfair differs between people (also within socioeconomic groups).

2. Financial insecurity is considered a confounder: it is not fully clear. I would consider it more likely a mediator or an alternative measure of SES. In any of those cases, controlling for financial insecurity might lead to overadjusted (underestimated) association of SES with the outcomes.

3. Are the Results of Study 2 not too elaborate? Reading the research questions, one is waiting until line 348 and onwards where the answers follow, particularly Figure 2. All the previous is very much secondary. The same holds for several of the additional outcomes. The main one in the UG is "number of acceptances of unfair offers". All others are secondary (evaluation of proposer and intention for future interaction). Try to concentrate on the main findings and help readers to see where the primary focus should be.

4. Table 3 indicates that SES (1 and 2) is related to perceived unfairness (3), but not to the number of acceptances of unfair offers (6). Figure 2, however, shows that there is an association between SES and the number of acceprantes of unfair offers (which is mediated by the perceived unfairness). This is really puzzling. If it is caused by some kind of suppressor effects, please clarify.

Minor comments

1. Political orientation (1 very liberal and 7 very conservative) is supposed to correlate with acceptance of unfair treatment. What is the hypothesised direction of that association (and why?)?

2. Study 2 are all students: is the variation in the key determinant, i.e. SES, not restricted by this selection?

3. The unfair offers in both Study 1 and Study 2 should be explained a bit further (preferably not using more words). In Figure 1 for example, it is unclear that the proposer gets 26 and the recipient 4 (experimenter announces) and then there are 3 rounds. It is just unclear, explain super-clearly (it is fascinating to know exactly how the unfairness is created).

4. When rejecting, both proposer and recipient get zero coins (line 263): would knowing that imply that one needs to control for additional variables (e.g. solidarity, emphathy)? Why are the participants informed like that?

5. Check carefully, I saw several instances of plural ("are") where it should be singular ("is) (e.g. line 424, 401).

Reviewer #2: The authors used two studies to explore the difference of higher and lower socioeconomic status (SES) during unfair treatments. The topic is potentially of interest, however, as detailed below, the manuscript in its current form has several structures, academic expression, experimental design, and statistics shortcomings that need to be addressed.

Introduction

1, Line 65, who are the other people (Kim et al., 2021)? Please clarified.

2, The relationship between perceive unfair treatments and tolerance for unfairness should be clarified. The authors addressed that “lower SES individuals—would perceive unfair treatments as less problematic than higher SES individuals (Line 80-81)”, and then the downstream consequences are differences in one’s tolerance for unfair treatments (Line 83-84). However, in Line 95-97, the authors illustrated that “…how SES differences in initial tolerance for unfairness could have downstream consequences…”. This sentence means that initial tolerance for unfairness is ealier than perceive unfair treatments. Likewise, in introduction, authors proposed that the first goal is to test the prediction “lower SES individuals would perceive unfair treatments as less problematic than higher SES individuals”. However, in “The Present Research” section, authors proposed that the first goal is to test “lower SES individuals would be more tolerant to unfair treatments than higher SES individuals”. Perceive unfair treatments and tolerance for unfairness are distinctive term.

3, the difference between DG and UG should be addressed with more details.

Study 1

1, in the Introduction, the authors identified one’s tolerance for unfair treatments as the likely to take an action against the unfair treatment (Line 85-86). However, in the study 1, the authors suggested that “participants indicated how unfair the offers from the proposer were (1 = not at all to 7 = a great deal; M = 5.75, SD = 1.34) as an index of tolerance for unfair treatments.” (Line 178-179). This measurment is related to “perceive unfair treatments”, rather than tolerance for unfair treatments. Moreover, the first sentence in Study 1 also mentioned that the aim of the Study 1 is to related to perceive unfair.

2, It is not clear whether SES groups mean the subjective SES categories. Moreover, 95% CI should be reported regarding the correlations. I suggested that the regression analysis is more suitable to reveal the relations between subjective SES classes and perceived unfairness, with gender, age, political orientation, system justification belief, and financial insecurity as covariations.

Study 2

1, Line 322, “indicating that lower SES individuals tended to be more tolerant to apparent unfair distributions during the DG” also confused the tolerant to apparent unfair distributions and perceived apparent unfair distributions.

Discussion

1, The authors can further illustrate that how their data contribute on the theories of fairness.

6. PLOS authors have the option to publish the peer review history of their article (what does this mean?). If published, this will include your full peer review and any attached files.

Reviewer #1: **Yes: **Hans Bosma

Reviewer #2: No

---

## [Author Response · Author response to Decision Letter 0]

23 Apr 2022

Please see the uploaded "response to reviewers" where we responded to each point raised by the editor and two reviewers.

---

## [Editor Report · Decision Letter 1]

27 Apr 2022

Socioeconomic status differences in psychological responses to unfair treatments:

Behavioral evidence of a vicious cycle

PONE-D-22-04413R1

Dear Dr. Na,

We’re pleased to inform you that your manuscript has been judged scientifically suitable for publication and will be formally accepted for publication once it meets all outstanding technical requirements.

On a personal note, I very much appreciate the work you put into this revision and the thoroughness of your response letter. Thank you for your efforts and congratulations! (Galak)

Kind regards,

Jeff Galak, PhD

Academic Editor

PLOS ONE
---

## [Editor Report · Acceptance letter]

1 Jun 2022

PONE-D-22-04413R1 

Socioeconomic status differences in psychological responses to unfair treatments: Behavioral evidence of a vicious cycle 

Dear Dr. Na:

I'm pleased to inform you that your manuscript has been deemed suitable for publication in PLOS ONE. Congratulations! Your manuscript is now with our production department. 

Kind regards, 

on behalf of

Dr. Jeff Galak 

Academic Editor

PLOS ONE